# Genotype–Phenotype Correlation in a Large Cohort of Eastern Sicilian Patients Affected by Phenylketonuria: Newborn Screening Program, Clinical Features, and Follow-Up

**DOI:** 10.3390/nu17030379

**Published:** 2025-01-21

**Authors:** Maria Chiara Consentino, Luisa La Spina, Concetta Meli, Marianna Messina, Manuela Lo Bianco, Annamaria Sapuppo, Maria Grazia Pappalardo, Riccardo Iacobacci, Alessia Arena, Michele Vecchio, Martino Ruggieri, Agata Polizzi, Andrea Domenico Praticò

**Affiliations:** 1Postgraduate Training Program in Pediatrics, University of Catania, 95125 Catania, Italy; 2Unit of Expanded Neonatal Screening and Inherited Metabolic Diseases, Department of Clinical and Experimental Medicine, AOU “Policlinico”, PO “G. Rodolico”, University of Catania, 95100 Catania, Italy; luisa.laspina@hotmail.it (L.L.S.); cmeli@policlinico.unict.it (C.M.); mmessina@unict.it (M.M.); doc.megpappalardo@gmail.com (M.G.P.); riki713@hotmail.it (R.I.); alessia.arena@gmail.com (A.A.); 3Unit of Pediatric Clinic, Department of Clinical and Experimental Medicine, University of Catania, 95123 Catania, Italy; lobiancomanuela@gmail.com (M.L.B.); agata.polizzi1@unict.it (A.P.); 4Unit of Pediatrics and Pediatric Emergency, “AOU “Policlinico”, PO “San Marco”, 95100 Catania, Italy; annamaria.pan@gmail.com; 5Rehabilitation Unit, Department of Biomedical and Biotechnological Sciences, University of Catania, 95123 Catania, Italy; michele.vecchio@unict.it; 6Unit of Pediatrics, Department of Medicine and Surgery, University Kore of Enna, 94100 Enna, Italy

**Keywords:** genetics, genotype–phenotype, mutations, phenylketonuria

## Abstract

**Background**: Phenylketonuria (PKU) is an autosomal recessive disorder caused by mutations in the phenylalanine hydroxylase (PAH) gene, leading to impaired amino acid metabolism. Early diagnosis through newborn screening (NBS) enables prompt treatment, preventing neurological complications. This study aims to describe the genetic and phenotypic spectrum of PKU and mild hyperphenylalaninemia (m-HPA) in patients diagnosed at the Department of Inborn Errors of Metabolism and Newborn Screening, Hospital G. Rodolico-S. Marco, Catania, over four decades (1987–2023). **Materials and Methods**: The retrospective analysis included 102 patients with elevated blood phenylalanine (Phe) levels born in Sicily and followed at the Institute. The phenotype evaluation comprised the Phe levels at birth/diagnosis, dietary tolerance, and sapropterin dihydrochloride responsiveness. The dietary compliance and Phe/Tyr ratios were assessed and compared across phenotypic classes and age groups. **Results**: Of 102 patients, 34 were classified as having classic PKU, 9 as having moderate PKU, 26 as having mild PKU, and 33 as having m-HPA, with a median age of 21.72 years. Common PAH variants included c.1066-11G>A (26/204 alleles), c.782G>A (18/204 alleles), and c.165delT (13/204 alleles). The phenotypes sometimes diverged from the genotype predictions, emphasizing dietary tolerance over the initial Phe levels for classification: m-HPA was statistically associated with a higher dietary tolerance (*p* < 0.001) compared to the classic, moderate, or mild forms of PKU. **Conclusions**: This study highlights the importance of large databases (e.g., BioPKU) for phenotype prediction and treatment optimization. Regular assessment of Phe/Tyr ratios is crucial for monitoring adherence and health. Phenotype determination, dietary management, and emerging therapies (Pegvaliase and gene therapy) are key to improving outcomes for PKU patients.

## 1. Introduction

Phenylketonuria (PKU), as Nenad Blau wisely defined it, can be considered “the book of Genesis” in the world of genetic disorders, being the first recognized biochemical cause of developmental delay and the second known inborn error of metabolism [1]. PKU is caused by mutations of the phenylalanine hydroxylase gene (PAH) and is the most common autosomal recessive Mendelian phenotype of amino acid metabolism: currently, 1723 PAH variants have been reported [2].

The PAH gene is located at chromosome 12q23.1 and consists of 13 exons and 12 introns. Tetrahydrobiopterin (BH4) is an indispensable cofactor that activates PAH, so defects involving the BH4 pathway may lead to a rise in Phe blood levels because of secondary PAH impairment [3]. PKU is present worldwide, affecting 0.45 million individuals, with a global prevalence of 1:23,930 live births; Italy, followed by Ireland, has been reported to have the highest prevalence of PAH deficiency (1:4000) out of 64 different countries [4]; we assume that Sicily has a prevalence consistent with Italy or even higher. Newborn screening (NBS) allows, nowadays, for the identification of all cases of PKU, facilitating the definition of its epidemiologic distribution [5,6].

According to Blau et al. [7], PAH deficiency may be classified into four different classes based on the highest untreated blood Phe concentration at newborn screening: classic (Phe > 1200 μmol/L), moderate (Phe 900–1200 μmol/L), mild (Phe 600–900 μmol/L), and mild hyperphenylalaninemia (MHPA, Phe < 600 μmol/L). Although these categories are useful to phenotype PAH deficiency, more recently, this classification has been replaced by ESPKU guidelines [8]: more specifically, based on dietary tolerance [9], it is possible to distinguish classic PKU (250–300 mg/day), moderate PKU (350–400 mg/day), mild PKU (400–600 mg/day), and MHPA in off-diet patients. This reclassification is very important, especially for those countries where newborn screening has been activated and where the early start of treatment usually precedes the Phe concentration peak.

Genetic analysis does not only play a diagnostic confirmation role in patients with PKU and MHPA but could also be performed to predict the biochemical and metabolic phenotype, sapropterin dihydrochloride cofactor responsiveness, and to give more complete information on the dietary tolerance, treatment options, and prognosis of affected patients and their parents [10,11]. The worldwide spread of PKU genetic analysis has contributed to enriching an extremely wide and heterogeneous landscape that today consists of over 1700 variants of the PAH gene [12].

The screening center in Catania began its work experimentally in 1987, collecting samples of all the newborns from the Sicilian provinces of Catania, Ragusa, and Syracuse through a project supported by the Italian Red Cross. In January 2011, a pilot project for extended newborn screening was started; since December 2017, following the Italian law that decreed mandatory and free newborn screening for all Italian newborns, two screening centers have been officially established in Sicily: one in Catania and one in Palermo, respectively, for Eastern Sicily (provinces of Catania, Messina, Ragusa, Siracusa, and Enna) and western Sicily (Palermo, Trapani, Agrigento, and Caltanissetta) [13]. Since 1987, our center has diagnosed 130 patients with PKU and MHPA throughout Eastern Sicily. For this study, 102 patients were selected based on the availability of complete medical records and adherence to follow-up protocols, ensuring a representative and comprehensive cohort. We excluded patients without genetic confirmation of their diagnosis, those who were deceased, individuals heterozygous for PAH variants, and cases with incomplete records on dietary tolerance.

The aim of our study is to provide a comprehensive description of the genetic and phenotypic spectrum of Sicilian patients diagnosed with phenylketonuria (PKU) or mild hyperphenylalaninemia (MHPA). This research encompasses a large cohort of patients who were identified and managed at the Department of Inborn Errors of Metabolism and Newborn Screening of the G. Rodolico-S. Marco Hospital in Catania. By analyzing medical records retrospectively from January 1987 to 2023, we aim to delineate the clinical characteristics, biochemical profiles, and genetic variants present in this population. Our study seeks to identify patterns in diagnosis, management, and outcomes over time. Furthermore, we intend to evaluate the impact of newborn screening programs on early detection and treatment, as well as the relevance of specific genetic mutations in disease expression. This work aspires to provide insights that can enhance the understanding of PKU and MHPA in a genetically defined context and contribute to optimizing care strategies for affected individuals.

This study was approved by the Ethical Committee of the University of Enna (254/2023, 2 September 2023). Written informed consent for publication was obtained from the participating parents of the patients.

## 2. Material and Methods

### 2.1. Population Study

Our retrospective analysis spans data collected from January 1987 to January 2023. The study population consists of all patients born in Sicily with increased blood phenylalanine (Phe) levels above 110 μmol/L, diagnosed through both biochemical and genetic analyses, and followed at our Institute. All patients originated from the eastern part of Sicily (population 2.5 million), reflecting the regional catchment area of our center: all the patients were diagnosed (by both biochemical and genetic analysis) and followed in our Institute. Prior to the implementation of the newborn screening (NBS) program in Sicily, diagnoses were based on clinical features. After the introduction of NBS, all patients were identified through screening. We excluded patients without genetic confirmation of diagnosis, those who were deceased, individuals heterozygous for PAH variants, and cases with incomplete records on dietary tolerance.

### 2.2. Diagnosis

Phe blood levels were determined by dried blood spots (DBSs) and bacterial inhibition assay for patients screened or diagnosed between 1987 and 2011 or by tandem mass spectrometry from 2011 to 2023. The collection of samples until 1992 was not homogeneous because it was not always performed in the standard 36–72 h. To exclude tetrahydrobiopterin deficiencies, we performed combined BH4 loading test with phenylalanine until 2013; subsequently, this method has been replaced by dosing pterins (blood and urine) and DHPR enzyme activity. After a pathological NBS finding or in case of an altered dosage of Phe in a symptomatic patient (values above 110 mmol/L), this was sent to our center to perform second-level diagnostic examinations (plasma amino acids, pterins, and calculation of the Phe/Tyr ratio on DBS) and genetic testing.

### 2.3. Phenotype Determination

All newborns with a positive screening suspicious for hyperphenylalaninemia were contacted and sent to the Clinical Center to receive a specific diagnosis. In case of PAH deficiency, newborns started specific dietetic treatment within one month of life. To characterize their phenotype, according to guidelines, patients were subjected to a period of washout from Phe for a variable period depending on the initial Phe value. After that, Phe was reintroduced, increasing the intake by 50 mg each week up to the maximum possible dietary tolerance. Tolerance of Phe has been defined as the amount of phenylalanine introduced in the diet by which patients are able to maintain blood levels of Phe below 360 μmol/L. Each patient was classified, according to Guldberg classification based on Phe dietary tolerance, as having classic PKU (250–300 mg/day), moderate PKU (350–400 mg/die), mild PKU (400–600 mg/die), or M-HPA (off-diet). Patients with a diagnosis made using NBS were also classified according to the Blau classification [4] based on Phe blood levels at the screening time, respectively, as having classic PKU (>1200 μmol/L), moderate PKU (900–1200 μmol/L), mild PKU (600–900 μmol/L), or m-HPA (<600 μmol/L).

### 2.4. Genotype Determination

PAH variants were determined by genetic laboratories (and Oasi Maria SS in Troina and Meyer’s Children Hospital in Florence) using the Sanger method or next-generation sequencing (NGS) on a comprehensive panel of 12 genes.

### 2.5. Phenotype–Genotype Correlation

For each genotype, we evaluated the corresponding phenotype, consisting of the following: Phe values at birth or at diagnosis, dietary tolerance, compliance to diet, and, where tested, possible response to sapropterin dihydrochloride cofactor. Furthermore, we analyzed dietetic compliance of our patients, and, where a large number of measurements were performed (at least 10 determinations every 30–90 days in two years), the Phe/Tyr ratios were calculated and compared within the various phenotypic classes and age groups. Where available, genotypes were searched for phenotype prediction and responsiveness to sapropterin dihydrochloride cofactor in the BioPKU database [2]. We compared the concordance between the classification of phenotypes based on values at birth (Blau) and that based on dietary tolerance (Guldberg).

With the aim of analyzing the predictive capacity of BioPKU, our phenotypes reported on BioPKU, including possible responsiveness to sapropterin dihydrochloride cofactor, were compared with the actual phenotype. To make this comparison, we considered mild and moderate classes (of Guldberg classification) as mild because BioPKU uses a three-class subdivision.

### 2.6. Data Collection

Data on our patients have been collected from electronic and/or medical records, from paper screening records and, more recently, from the screening management software of the screening center of Catania (SGService—Screening Center^®^, Scientific Gear Service, Zhunan, Taiwan).

### 2.7. Statistical Analysis

Statistical analysis was performed by using Excel in Microsoft^®^ Office 2016 (Microsoft Corporation, Redmond, WA, USA) and the software Jamovi version 2.3.28 (Sydney, Australia). Using these platforms, we calculated *p* Values and Pearson’s correlation coefficient between levels of Phe in blood at birth between classes based on dietary tolerance. Independent *t*-tests were performed to evaluate significant differences in Phe/Tyr ratio among different phenotypic classes, among patients with classical PKU adherent and non-adherent to diet, and among patients with classical PKU at preschool (age under 6 years) or school/adult age. We considered test results significant when *p* was <0.05.

## 3. Results

### 3.1. Overview

We recruited 422 patients: 227 patients with PKU and 195 patients with M-HPA (Figure 1). Among these, 320 patients did not meet the inclusion criteria and were excluded (311 patients had no available genetics, 5 patients had only one pathogenic allelic variant of PAH, and 4 patients died). A total of 102 patients with PKU or M-HPA met our inclusion criteria. Among these, 49/102 were female and 53/102 were male (male-to-female ratio 1:1.08), with an actual median age of 21.72 years.

A total of 10 patients (1 female, 9 male) were diagnosed because of symptoms; 92 out of 102 patients were diagnosed via newborn screening. The mean values of Phe at birth of the patients diagnosed using NBS are reported in Table 1.

According to Phe diet tolerance values, we identified 34/102 classic PKU cases, 9/102 moderate PKU cases, 26/102 mild PKU cases, and 33/102 mild hyperphenylalaninemia cases. A comparison (t-test) between the mean values of Phe at birth (92 out of 102 patients) for each pair of classes is reported in Table 2. The comparisons between the classic cases and mild cases, classic cases and M-HPA cases, moderate cases and M-HPA cases, and mild cases and M-HPA cases showed significant differences in the Phe values at birth.

### 3.2. Genetic Results and Variants Spectrum

All 102 patients underwent genetic analysis (Sanger and NGS): 17 patients were homozygous for PAH variants, while 85 were compound heterozygous. Among 204 alleles, we detected 42 pathogenic variants: 31 missense, 4 nonsense, 4 splice site variants, 2 frameshifts, and 1 in-frame deletion. The most common PAH variant reported in our population was c.1066-11G>A/IVS10-11G>A (26/204 alleles), followed by c.782G>A/R261Q (18/204 alleles) and c.165delT/p.F55Lfs*6 (13/204 alleles). Exons 6 and 7, coding for the central part of the PAH catalytic domain, were the most frequently affected sites with variants (16 out of 42 variants).

A total of 17 of the 102 (16.67%) patients inherited homozygous PAH variants relating to classic PKU (8/17), moderate PKU (1/17), mild PKU (3/17), and M-HPA (4/17). The other 85/102 patients (83.33%) showed compound heterozygous inheritance, resulting in the following phenotypes: classic PKU (25/85), moderate PKU (9/85), mild PKU (22/85), and M-HPA (29/85). The genotypes, phenotypes, and allelic distribution in the different phenotypes are reported in Table 3, Table 4 and Table 5.

### 3.3. Sapropterin Dihydrochloride Cofactor (Kuvan) Responsiveness

Of the 102 patients, 31 were tested for sapropterin dihydrochloride cofactor responsiveness; 7 (22.58%) responded well to treatment and switched to a free diet, while 1 patient was a partial responder. Responsiveness was defined as an increase in the amount of natural protein by 100% or more or improved biochemical control (phenylalanine > 75% in target range), demonstrated from a trial (up to 6 months) of treatment with BH4. All responders were classified as mild based on their Phe dietary tolerance, and the partially responsive patient was classified as having classic PKU. When compared with the BioPKU predictions, 4/7 patients had a correct prediction, with a probability of response ranging from 50% to 100%. The genotypes of the remaining three responders were not reported in BioPKU.

Among the 23/31 patients who were not responsive to sapropterin dihydrochloride cofactor, 14 were classified as having classic PKU, 4 as having moderate PKU, and 6 as having mild PKU. The genotypes of the patients who responded to Kuvan are reported in Table 6.

### 3.4. Follow-Up, Compliance with Diet, and Phe/Tyr Ratio

We evaluated the dietary adherence of all our patients with PKU: 17/69 patients with PKU showed no adherence to the diet (2 poor-compliant, 1 non-compliant, 1 who suspended the diet after pregnancy, 2 compliant only during pregnancy, and 11 who suspended the diet around the age of 16–20). Among these 17 non-compliant patients, 14 had a classic phenotype, 2 had a moderate phenotype, and 1 had a mild phenotype.

We collected the Phe/Tyr ratio values of the PKU patients adherent to the diet who had performed at least 10 determinations every 30–90 days over two years (12 classic PKU cases, 9 moderate PKU cases, and 12 mild PKU cases). The descriptive results and box plots are described in Table 7. The t-tests showed no significant differences in the Phe/Tyr ratios among different phenotypic classes (Appendix A). Analyzing the Phe/Tyr values in the patients with classic PKU, those non-compliant with the diet had significantly higher values than the compliant patients. Additionally, the school-aged and adult PKU patients had significantly higher Phe/Tyr ratios than the preschool-aged PKU patients (Appendix A, Figure 2).

### 3.5. Blau’s Classification/Guldberg’s Classification Concordance

We compared the classification based on the Phe values at birth with the classification based on the dietary tolerance in 92/102 patients; 10 patients were not examined because the diagnosis was not made at birth. The concordance between the two classifications was determined in 56/92 patients (14/56 classic, 1/56 moderate, 8 mild, and 33/56 m-HPA); in 36/92 patients, there was no concordance: 26/36 were underestimated by the classification based on the Phe values at birth (13/36 cases of one class, in 7/36 cases of two classes, and in 6/36 cases of three classes); and 10/36 patients were overestimated (in 7/36 cases of one class and in 3/36 cases of two classes).

We finally compared the BioPKU phenotype prediction with the actual phenotype in 81/102 patients (21/102 of our genotypes were not available on BioPKU). We considered the mild and moderate classes (of dietary tolerance classification) as mild because BioPKU uses a three-class subdivision. Among these, in 63/81 cases, we found concordance between the database prediction and the real phenotype of the patient (25/34 classic, 18/30 mild/moderate, and 20/21 m-HPA). In 17/61 cases, there was no concordance between the prediction and the actual phenotype; in 12 cases, the BioPKU prediction overestimated the phenotypic class (in 5/12 cases of one class, in 6/12 cases of two classes, and in 1/12 cases of three classes); and in 5 cases, it was underestimated (in 4/5 cases of one class and in 1/5 cases of two classes).

### 3.6. Pearson’s Correlation Coefficient

Among our patients, the Phe levels at birth and Phe dietary tolerance (not considering m-HPA) were negatively correlated: r(58) = −0.35, *p* < 0.001 (Figure 3).

## 4. Discussion

Italy, followed by Ireland, has been reported to have the highest prevalence of PAH deficiency (1:4000) out of 64 different countries [1]. We assume that Sicily has a prevalence of hyperphenylalaninemias, consistent with Italy, or even higher.

Researching hyperphenylalaninemias using NBS has significantly altered the natural history of these diseases, in which early diagnosis and treatment in the first months of life are essential to avoid long-term neurological sequelae. Our retrospective study offers a precise overview of PKU and m-HPA in a large cohort of patients from Eastern Sicily, as all our patients have been diagnosed, treated, and closely followed-up with over decades at the same Clinical Center. We focused on the importance of defining the genetic and phenotypic framework of these patients, with the aim of establishing their dietary tolerance as early as possible. As widely reported and expected, 83.33% of our patients are compound heterozygous, slightly higher compared to the rate in the worldwide population (73%) [1]. Most of the variants found were missense (73%), with a fair amount of splice site and nonsense variants, and frameshifts and deletions variants being less represented. The most common variant detected in our population was c.1066-11G>A/IVS10-11G>A (13%), which is the second most common variant reported in Europe [1,13]; R408W/c.1222C>T, the most reported variant in Europe and worldwide, associated with almost no residual activity of PAH, was detected only in one patient. The great variability of the genotypes has not allowed for the identification of a significant group of more frequent genotypes, i.e., the genotype c.1066-11G>A/c.1243G>A is the most detected, but it is present in only three cases: two of m-HPA and one of mild PKU. Exon 6 and 7, coding for the central part of the PAH catalytic domain, were the most involved sites of variants (38%), and this is consistent with the literature [5]. Overall, the patients with homozygous inheritance showed a prevalence of more severe phenotypes, with 50% (8/16 pt) having classical phenotypes: these data are comparable with data from Southern Europe [5], while the global incidence of classic PKU is higher than that of m-HPA and the mild phenotype.

A very large collection of genetic results from the French population showed a comparable high incidence of the intronic mutation c.1066-11 G>A: it was the most common in both the populations examined (French and Sicilian). The mutation c.1208 C>T, found to be significantly enriched in a prespecified geographical area (Mediterranean France) was also among the most commonly found in our population (11 patients), thus suggesting a similar genetic origin dating back to early Mediterranean people or Greek/Roman ancestry [13]. The other specific regional “common” mutation (North, Northwest, and Africa) from the French study were not found in our patients.

Establishing the phenotype and dietary tolerance of patients with hyperphenylalaninemia early is critical for setting the appropriate dietary regimen, assessing the BH4 responsiveness, and identifying patients at a higher risk of diet abandonment, who require closer monitoring and support. Similar studies emphasize the importance of early phenotype determination to guide treatment strategies and long-term adherence to care [6,7,14,15]. In our cohort, 30% (31/102) of the patients were tested for BH4 responsiveness. Seven (23%), all with mild PKU, responded well and transitioned to a free diet, while one patient with classic PKU showed partial responsiveness. The genotypes of the responders listed in the BioPKU database were correctly predicted with a 50–100% probability of response. Conversely, 74% (23/31) did not respond, with 14 correctly predicted as non-responders based on the genotype, while 5 were improperly predicted as being responsive (4 lacked entries in BioPKU). As noted in the literature, the BioPKU database is more effective for deciding whether to test a patient than for accurately predicting the treatment response [14,15].

Although, in the literature, it has been described that the variants involved in protein oligomerization appear to be those with a greater responsiveness to BH4, of ten different variants reported in our patients, only one variant (c.1241A>G) is in the oligomeric binding domain, while all other variants affect the catalytic domain [5].

Non-adherence emerged in 25.49% of our patients, particularly after reaching adulthood. Most of these patients (80.77%) had a classic phenotype requiring restrictive diets based on artificial protein blends. Historically, scientific guidelines suggested discontinuing the diet after age 18, an expectation later abandoned due to the risks of long-term neurological and psychiatric complications [15]. Among the non-adherent patients, three resumed the diet exclusively during pregnancy to mitigate fetal risks from maternal PKU.

In evaluating the adherence to the diet of our patients, we also focused on the role that the Phe/Tyr ratio might play in their follow-up. Our study shows that, if a good adherence to the diet is established, there are no substantial differences in the Phe/Tyr ratio between the phenotypic classes; the difference becomes significant when patients abandon the diet or become older. Our data, combined with our extensive clinical experience, testify how patients at the start of school and the increase in sociality, especially in adolescence, significantly reduce their adherence to the PKU diet. In our opinion, continuously monitoring the Phe/Tyr ratio is a crucial parameter for assessing adherence and enabling timely intervention when necessary. For these reasons, starting from 2011, thanks also to the possibility of exploiting mass spectrometry, we studied and patented at our center a PKU Smart sensor to simplify and improve the follow-up of our patients. It is a reliable and miniaturized point-of-care analytical method, by which patients can measure the Phe, Tyr, and Phe/Tyr ratio at home with a single drop of blood; the data are instantly sent to the physician who can then monitor in real time the biochemical trends in the patient [16,17,18,19].

Delineating a precise picture of the genotype–phenotype correlation of PKU is hampered by some factors: the existence of numerous mutations and even greater variability in the combination of them, the heterogeneity of the methods and criteria used for diagnosis and classification, and the biochemical mechanisms of interaction between two mutated polypeptides. In addition to these factors, the influence of gene modifiers has been widely described, which would be able to affect the metabolism of phenylalanine and, consequently, the degree of severity of the phenotypic manifestation [6,9]. In 2019, Garbade et al., demonstrating a better agreement than previous studies [8], developed a new phenotype–genotype correlation model tested on 9336 patients with PKU, who were assigned an allelic phenotypic value (AVP) based on their biochemical phenotype (0 = cPKU, 5 = mPKU, and 10 = MHP). The variants with the AVP with the highest score were therefore considered dominant and constituted the genotype–phenotype values (GPVs).

The importance of considering allele interactions is highlighted by an observed difference in the assigned phenotype in some variants, depending upon whether they are caused by a homozygous or a heterozygous genotype. The GVPs are currently available on BioPKU [4], which has been consulted for all the variants of our database, where they are reported. More specifically, 77.04% of our patients reported in the database displayed concordance with the prediction and the phenotype based on their dietary tolerance. Notably, 27.87% (17/61 pt) of our patients were not correctly predicted by BioPKU, which overestimated 70.59% (12/17 pt) of them and underestimated the remaining 30%. Consequently, our study allows us to outline phenotypic classes that are different from those expected and predicted on the basis of their genotype from BioPKU. However, although we can consider these results as showing good concordance, a different response was found when comparing the Phe levels at the first NBS: Pearson’s correlation coefficient between the Phe at NBS and the Phe dietary tolerance, although technically showing a negative correlation, demonstrated only a weak relationship. The concordance between the Blau classification at birth and dietary tolerance was in fact determined only in 61.29% of patients (57 pt); interestingly the phenotype was mainly underestimated in 27.96%, even of three classes. Conversely, 100% of the cases with m-HPA were correctly classified based on the Phe values at birth [5,18].

In summary, the Phe blood levels at birth might be accurate for the extreme phenotypes, especially for m-HPA, but such patients are subjected to misdiagnosis when their phenotype is intermediate. As recommended by ESPKU guidelines, our data confirm that classifying PAH deficiency based on the Phe levels at birth should be discontinued, and the primary objective of the clinician must be to establish tolerance promptly [17,18]. One of the major challenges in PKU management is predicting the severity and progression of the disease based on genetic mutations alone. However, we can assume that BioPKU is a good predictor of the phenotype but shows lower accuracy for predicting the BH4 responsiveness [5].

### Study Limitations

Despite the strengths of our study, several limitations should be acknowledged. First, the retrospective design may have introduced selection bias or incomplete data collection. Second, the cohort is limited to a single-center population from Eastern Sicily, which may not fully represent the genetic and phenotypic variability in other regions. Third, while the BioPKU predictions provided valuable insights, they may not account for rare genetic variants or environmental factors influencing the phenotype. Finally, the lack of long-term follow-up data on newer therapeutic approaches, such as Pegvaliase or gene therapy, limits our ability to assess their broader clinical impact.

## 5. Conclusions

Our retrospective study was conducted on a large cohort of patients from Eastern Sicily, all diagnosed and followed at a single center with over thirty years of expertise in managing inborn errors of metabolism, including phenylketonuria (PKU). This long-term experience highlights the pivotal role of centralized, specialized care in improving outcomes for rare metabolic disorders. One of the key aspects underscored in our study is the utility of leveraging comprehensive databases, such as BioPKU, which provide accessible resources for physicians to predict potential phenotypes based on identified genotypes and guide decisions regarding BH4 responsiveness testing [2,3]. However, it is crucial to acknowledge the complexity of phenotype prediction, given the significant genetic variability, allelic interactions, and the influence of modifier genes that often complicate precise prognostication [4,20,21].

Our findings demonstrate that patient phenotypes frequently deviate from database predictions, reinforcing the importance of clinical assessments beyond the initial genetic findings. Notably, our study emphasizes that dietary tolerance, rather than neonatal Phe levels, serves as a more accurate determinant of the clinical phenotype. This aligns with previous reports suggesting that long-term metabolic control, dietary compliance, and individualized tolerance thresholds are better predictors of outcomes [6,19]. Specifically, for patients diagnosed with classical PKU, adherence to a strict lifelong diet remains essential but challenging, as they represent the group most vulnerable to treatment abandonment and its associated neurological and cognitive complications [7]. These individuals require closer monitoring and tailored interventions to mitigate risks.

We also stress the importance of periodically calculating the Phe/Tyr ratio as a biomarker for metabolic control. This metric, combined with advances in monitoring tools, has been validated as a reliable indicator of treatment adherence and metabolic status, enabling clinicians to intervene promptly to safeguard patient health [22,23].

Establishing an accurate clinical phenotype through the integration of genetic, biochemical, and dietary data allows healthcare professionals to optimize treatment strategies. This includes exploring alternative or adjunctive therapies, such as Pegvaliase or emerging gene therapy approaches, particularly for patients with more severe phenotypes that remain poorly controlled on conventional treatment regimens [8,24,25]. With the advent of personalized medicine, incorporating new therapeutic tools and continuous patient-centered care can significantly improve long-term outcomes for individuals with PKU [25,26].

## Figures and Tables

**Figure 1 nutrients-17-00379-f001:**
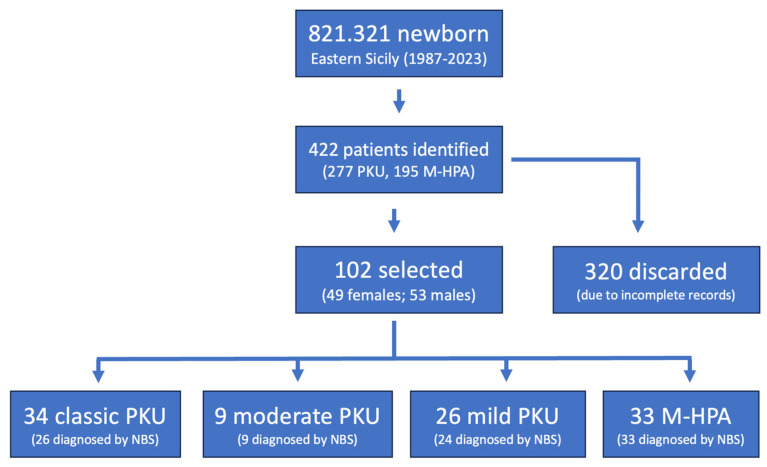
Patient selection flowchart. Legend: M-HPA: Mild hyperphenylalaninemia; NBS: newborn screening; PKU: phenylketonuria. Of the excluded patients, 311 had no available genetics; 5 had one pathogenic variant of PAH; 4 died.

**Figure 2 nutrients-17-00379-f002:**
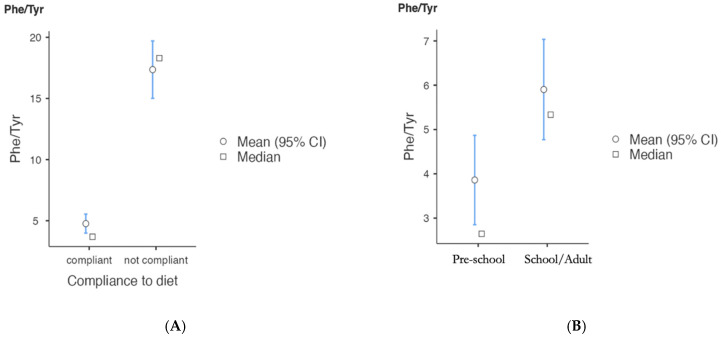
Plots on adherence to diet compared between compliant and not-compliant classic PKU patients (*n* = 12 compliant; *n* = 14 non-compliant) (**A**) and based on age in the compliant patients (*n* = 12) (**B)**. Legend: Pre-school: <6 years of age; school/adult: 6–60 years.

**Figure 3 nutrients-17-00379-f003:**
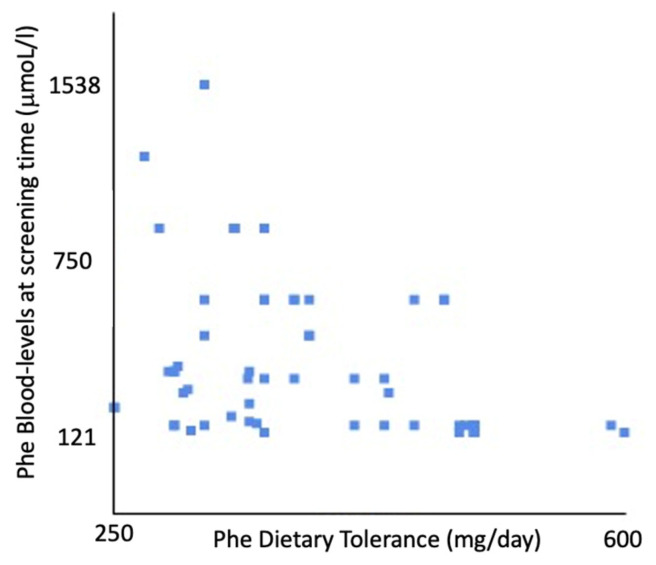
Pearson’s correlation coefficient (0.35). On the X axis are levels of Phe in blood at the screening time; on Y axis are levels of Phe dietary tolerance. We examined 92 patients.

**Table 1 nutrients-17-00379-t001:** Mean value of Phe at birth in each class of Guldberg’s classification in our patients (10/102 patients were excluded because diagnosis was made based on symptoms).

	Classic PKU	Moderate PKU	Mild PKU	m-HPA
N° of patients	26	9	24	33
Mean value of Phe at birth	1138.21 mmol/L	766.17 mmol/L	682.43 mmol/L	233.19 mmol/L
Standard deviation	588.83	333.33	305.54	71.22

**Table 2 nutrients-17-00379-t002:** T-test. Comparison between each pair of classes (based on dietary tolerance) and the mean values of Phe in NBS.

Classes (Dietary Tolerance)	*p* Value
Classic vs. Moderate	0.076
Classic vs. Mild	0.002
Classic vs. m-HPA	<0.001
Moderate vs. Mild	0.614
Moderate vs. m-HPA	<0.001
Mild vs. m-HPA	<0.001

**Table 3 nutrients-17-00379-t003:** Allelic frequency for each class of tolerance classification.

Allele	Classic PKU	Moderate PKU	Mild PKU	m-HPA	TOT	Classic PKU%	Moderate PKU%	Mild PKU%	m-HPA%
c.1028A>G	3				3	4.41%			
c.1045T>C	4		1		5	5.88%		1.92%	
c.1066-11G>A	8	3	9	6	26	11.76%	16.67%	17.31%	9.09%
c.1139C>T				1	1				1.51%
c.1157A>G			1		1			1.92%	
c.1169A>G				1	1				1.51%
c.1181A>C				1	1				1.51%
c.1184C>G			1		1			1.92%	
c.1208C>T				11	11				17.2%
c.121C>T		1			1		5.56%		
c.1222C>T		1			1		5.56%		
c.1223G>A	1		4		5	1.47%		7.69%	
c.1241A>G	1	1	2	1	5	1.47%	5.56%	3.85%	1.51%
c.1241A>T		1			1		5.56%		
c.1243G>A			1	2	3			1.92%	3.03%
c.1315+1G>A			1		1			1.92%	
c.143T>C	5	1	4	3	11	7.35%	5.56%	7.69%	4.54%
c.165delT	7	2	1	3	13	10.29%	11.11%	1.92%	4.54%
c.165T>G				1	1				1.51%
c.168+5G>C	3				3	4.41%			
c.194T>A	2				2	2.94%			
c.283_285delATC			4		4			7.69%	
c.331C>T	3		1		4	4.41%		1.92%	
c.441+5G>T			1		1			1.92%	
c.434A>T				1	1				1.51%
c.473G>A	3	1	2	2	8	4.41%	5.56%	3.85%	3.03%
c.533A>G				1	1				1.51%
c.526C>T	5	1		1	7	7.35%	5.56%		1.51%
c.561G>A	4			1	5	5.88%			1.51%
c.581T>C	1				1	1.47%			
c.592_613del	1				1	1.47%			
c.601C>T				1	1				1.51%
c.631C>A			1	4	5			1.92%	6.06%
c.638T>C			1		1			1.92%	
c.688G>A				3	3				4.54%
c.754C>T	1			1	2	1.47%			1.51%
c.721C>T				1	1				1.51%
c.776C>T	1				1	1.475%			
c.781C>T	6	3			9	8.82%	16.67%		
c.782G>A	5	3	4	6	18	7.35%	16.67%	7.69%	11.11%
c.826A>G			1		1			1.92%	
c.829T>G			4		4			7.69%	
c.833C>A			2		2			3.85%	
c.842+5G>A				1	1				1.51%
c.842C>T	5		2	2	9	7.35%		3.85%	3.03%
c.848T>A			1		1			1.92%	
c.898G>T			2	9	11			3.85%	14.1%
c.926C>T			2		2			3.85%	
Tot	69	18	53	64	204				

**Table 4 nutrients-17-00379-t004:** Genotypes and phenotypes of patients with PKU. NA = not available.

N pt	Genotype		BioPKU	Phe at Birth/at Diagnosis (µmol/L))	Tolerance (mg/die)	Compliance
1	c.638T>C	c.638T>C	NA	908	500	compliant
2	c.473G>A	c.473G>A	Classic	430	235	compliant
3	c.526C>T	c.526C>T	Classic	1513	250	not compliant
4	c.526C>T	c.526C>T	Classic	1513	250	diet suspended at 17 years
5	c.1066-11G>A	c.1066-11G>A	Classic (74%)—Mild (26%)	696	255	compliant
6	c.1241A>G	c.1241A>G	Mild	420	350	compliant
7	c.1066-11G>A	c.1066-11G>A	Mild	1211	250	compliant
8	c.561G>A	c.561G>A	Classic	1513	230	diet suspended at 18 years
9	c.842C>T	c.842C>T	Classic	848	600	compliant
10	c.842C>T	c.842C>T	Classic	848	600	compliant
11	c.1066-11G>A	c.1066-11G>A	Classic	1090	380	compliant
12	c.1066-11G>A	c.1066-11G>A	Classic	1211	380	compliant
13	c.121C>T	c.121C>T	Mild	848	380	compliant
14	c.331C>T	c.331C>T	Classic	726	230	diet suspended at 20 years
15	c.1243G>A	c.1243G>A	MHPA	484	1200	diet only during pregnancy
16	c.165delT	c.165delT	Classical (73%)—Mild (26%)	726	600	compliant
17	c.782G>A	c.782G>A	NA	363	400	compliant
18	c.782G>A	c.782G>A	NA	363	400	compliant
19	c.1028A>G	c.1028A>G	Mild	1453	600	compliant
20	c.1028A>G	c.1028A>G	Mild	1453	free diet with BH4	compliant
21	c.165delT	c.165delT	NA	593	275	compliant
22	c.168+5G>C	c.168+5G>C	Classic (77%)—Mild (23%)	>1500	230	diet suspended at 16 years, compliant during pregnancy
23	c.781C>T	c.781C>T	Classic (83%)—Mild (17%)	>1500	250	compliant
24	c.781C>T	c.781C>T	Classic (83%)—Mild (17%)	>1500	250	compliant
25	c.473G>A	c.473G>A	Classic	>1500	230	diet suspended at 16 years
26	c.165delT	c.165delT	Classic (73%)—Mild (27%)	NA	250	poor compliant
27	c.165delT	c.165delT	Classic (73%)—Mild (27%)	>1200	250	poor compliant
28	c.898G>T	c.898G>T	MHPA	302	free diet with BH4	compliant
29	c.898G>T	c.898G>T	MHPA	494	free diet with BH4	compliant
30	c.1066-11G>A	c.1066-11G>A	Classic	NA	400	compliant
31	c.781C>T	c.781C>T	Mild	666	310	compliant
32	c.143T>C	c.143T>C	Classic (32%)—MHPA (1%)—Mild (67%)	242	free diet with BH4	compliant
33	c.143T>C	c.143T>C	Classic (34%)—Mild (66%)	>1200	600	compliant
34	c.1066-11G>A	c.1066-11G>A	Classic	378	415	compliant
35	c.143T>C	c.143T>C	Classic (75%)—Mild (25%)	2179	230	diet suspended at 18 years
36	c.473G>A	c.473G>A	Classic	1229	340	compliant
37	c.826A>G	c.826A>G	NA	908	free diet with BH4	compliant
38	c.592_613del	c.592_613del	NA	2126	250	N/A
39	c.165delT	c.165delT	NA	363	250	compliant
40	c.165delT	c.165delT	NA	NA	250	diet suspended at 18 years
41	c.165delT	c.165delT	NA	NA	250	diet suspended at 18 years
42	c.473G>A	c.473G>A	Classic	>1200	500	compliant
43	c.143T>C	c.143T>C	Mild	605	free diet with BH4	compliant
44	c.194T>A	c.194T>A	Classic	1392	250	compliant
45	c.842C>T	c.842C>T	Classic (99%)—Mild (1%)	121	300	compliant
46	c.842C>T	c.842C>T	Classic (99%)—Mild (1%)	121	300	compliant
47	c.1066-11G>A	c.1066-11G>A	Classic	1090	250	compliant
48	c.165delT	c.165delT	Classic	>1500	230	diet suspended at 18 years
49	c.143T>C	c.143T>C	Mild (63%), classic (36%), MHPA (1%)	1089	600 mg with BH4	compliant
50	c.1045T>C	c.1045T>C	Classic	908	500	diet suspended at 18 years
51	c.1066-11G>A	c.1066-11G>A	Classic	1513	250	diet suspended at 18 years
52	c.1241A>G	c.1241A>G	Mild	605	800	compliant
53	c.526C>T	c.526C>T	NA	660	300	compliant
54	c.526C>T	c.526C>T	NA	363	380	compliant
55	c.561G>A	c.561G>A	NA	1538	250	compliant
56	c.561G>A	c.561G>A	NA	484	250	compliant
57	c.1066-11G>A	c.1066-11G>A	Mild	484	500	compliant
58	c.782G>A	c.782G>A	Classic (32%)—Mild (68%)	726	380	compliant
59	c.781C>T	c.781C>T	Classic	>1500	250	N/A
60	c.1315+1G>A	c.1315+1G>A	Mild	666	400	compliant
61	c.168+5G>C	c.168+5G>C	Classic (1 case Mild)	1211	250	compliant
62	c.781C>T	c.781C>T	Classic	>1500	230	compliant
63	c.842C>T	c.842C>T	NA	666	260	compliant
64	c.143T>C	c.143T>C	Classic(44%)—Mild (55%)	339	400	compliant
65	c.1066-11G>A	c.1066-11G>A	Classic (44%)—Mild (56%)	339	340	compliant
66	c.1066-11G>A	c.1066-11G>A	Classic (73%)—Mild	1513	250	diet only during pregnancy
67	c.631C>A	c.631C>A	MHPA	605	free diet with BH4	compliant
68	c.1223G>A	c.1223G>A	Mild	726	600	diet suspended at 18 years
69	c.926C>T	c.926C>T	Classic (20%)—Mild (80%)	726	800	compliant

**Table 5 nutrients-17-00379-t005:** Genotypes and phenotypes of patients with m-HPA. NA = not available.

N pt	Genotype		BioPKU	Phe at Birth/at Diagnosis (µmol/L))	Tolerance (mg/die)
1	c.1066-11G>A	c.1066-11G>A	MHPA	333	free diet
2	c.842C>T	c.842C>T	MHPA (88%)—Mild (12%)	236	free diet
3	c.165delT	c.165delT	MHPA	314	free diet
4	c.898G>T	c.898G>T	MHPA	363	free diet
5	c.1208C>T	c.1208C>T	MHPA (96%)—Mild (4%)	303	free diet
6	c.631C>A	c.631C>A	NA	206	free diet
7	c.1208C>T	c.1208C>T	NA	163	free diet
8	c.1066-11G>A	c.1066-11G>A	MHPA	242	free diet
9	c.1208C>T	c.1208C>T	MHPA	182	free diet
10	c.1208C>T	c.1208C>T	NA	212	free diet
11	c.898G>T	c.898G>T	MHPA	363	free diet
12	c.1208C>T	c.1208C>T	MHPA	182	free diet
13	c.1066-11G>A	c.1066-11G>A	Classic	242	free diet
14	c.165delT	c.165delT	NA	242	free diet
15	c.1139C>T	c.1139C>T	NA	163	free diet
16	c.165delT	c.165delT	MHPA	363	free diet
17	c.782G>A	c.782G>A	NA	150	free diet
18	c.533A>G	c.533A>G	MHPA	182	free diet
19	c.782G>A	c.782G>A	MHPA	242	free diet
20	c.1066-11G>A	c.1066-11G>A	MHPA (82%)—Mild (18%)	182	free diet
21	c.898G>T	c.898G>T	MHPA	242	free diet
22	c.898G>T	c.898G>T	MHPA	242	free diet
23	c.842C>T	c.842C>T	MHPA (86%)—Mild (14%)	266	free diet
24	c.434A>T	c.434A>T	MHPA	127	free diet
25	c.1181A>C	c.1181A>C	MHPA	242	free diet
26	c.283_285delATC	c.283_285delATC	Mild	272	free diet
27	c.283_285delATC	c.283_285delATC	Mild	242	free diet
28	c.688G>A	c.688G>A	NA	115	free diet
29	c.473G>A	c.473G>A	MHPA	103	free diet
30	c.898G>T	c.898G>T	MHPA	303	free diet
31	c.165T>G	c.165T>G	MHPA (67%)—Mild (33%)	115	free diet
32	c.526C>T	c.526C>T	MHPA	163	free diet
33	c.754C>T	c.754C>T	MHPA (63%)—Mild (37%)	303	free diet

**Table 6 nutrients-17-00379-t006:** Genotypes and prediction of patients responsive to BH4. N/A = not available.

Pt	Allele/Protein Variant 1	Allele/Protein Variant 1	Responsiveness to BH4	BioPKU Prediction
1	c.1028A>G/Y343C	c.829T>G/Y277D	complete	yes (50%)—no (50%)
2	c.898G>T/A300S	c.1223G>A/R408Q	complete	N/A
3	c.898G>T/A300S	c.1223G>A/R408Q	complete	N/A
4 *	c.143T>C/L48S	c.143T>C/L48S	complete	yes (74%)—no (17%)—slow (9%)
5	c.826A>G/M276V	c.1066-11G>A/IVS10-11G>A	complete	N/A
6	c.143T>C/L48S	c.1241A>G/Y414C	complete	yes
7 *	c.143T>C/L48S	c.143T>C/L48S	partial	yes (75%)—no (17%)—slow (8%)
8	c.631C>A/P211T	c.782G>A/R261Q	complete	yes

* While both patients share the L48S variant, differences in their overall genetic, epigenetic, or metabolic backgrounds likely account for the variation in response. Identifying these factors could involve deeper genomic analysis or assessments of BH4 metabolism and PAH function in both individuals.

**Table 7 nutrients-17-00379-t007:** Phe/Tyr ratio among PKU phenotypic classes.

	Phenotype	Phe/Tyr
N	Classic	87
	Mild	108
	Moderate	85
Mean	Classic	4.35
	Mild	4.34
	Moderate	4.01
Median	Classic	3.55
	Mild	4.29
	Moderate	3.46
Standard Deviation	Classic	3.08
	Mild	2.04
	Moderate	3.04
Minimum	Classic	0.100
	Mild	0.310
	Moderate	0.110
Maximum	Classic	11.7
	Mild	9.73
	Moderate	12

## Data Availability

The data supporting the reported results can be found at the University Hospital “Policlinico-San Marco”, Catania, Italy.

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
