# Peer review of "Genotype–Phenotype Correlation in a Large Cohort of Eastern Sicilian Patients Affected by Phenylketonuria: Newborn Screening Program, Clinical Features, and Follow-Up"

_nutrients, 2025, doi:10.3390/nu17030379_

Round 1

Reviewer 1 Report

Comments and Suggestions for Authors

The manuscript refers to a retrospective analysis of the incidence and mutations of Phenylketonuria in Eastern Sicily. The article is interesting; the rationale is adequate. The techniques used are the standard for genetic diseases. However, the title of the manuscript is too long and confusing, it must be shortened, a suggestion Genotype-phenotype correlation in a large cohort of eastern Sicilian patients: Newborn Screening program, clinical features, and follow-up. The abstract has a similar problem; it must be concise yet informative. Due to the admixed population in Sicily, this reference should be helpful to discuss  and it will interesting to correlate the findings

  • DOI: 10.1016/j.ebiom.2019.102623
  • Please add a Table with the patient's clinical characteristics so it is easier to follow up. In Table 6, please discuss why the responsiveness of BH4 is partial in patient 7 compared to patient 4. Table 7 should be only the right side of the Figure. Table 8 is confusing; please modify to the short part and add a legend if necessary. Figure 1: how many individuals are analyzed? The Figure can be simplified by adding age. In Figure 2, please label the axis and add the correlation value even though none exists. The discussion should be analyzing the data and comparing it with other studies, not repeating the results and the conclusions, and adding the study's limitations separately.
  •  
  •  

Author Response

To Prof. Maria Luz Fernandez,

Department of Nutritional Sciences,

University of Connecticut, Storrs, USA

Editor-in-chief

Nutrients

Dear Editor,

We sincerely appreciate the opportunity to revise and resubmit our manuscript (now) entitled Genotype-phenotype correlation in a large cohort of eastern Sicilian patients: Newborn Screening program, clinical features, and follow-up” to Nutrients. We are grateful to the reviewers for their thoughtful and constructive comments, which have greatly helped us improve the quality and clarity of our work.

In this revised version, we have carefully addressed all the points raised by the reviewers and have incorporated their suggestions into the manuscript. Below, we provide a detailed, point-by-point response to each comment, specifying the changes made and referencing the relevant sections of the revised manuscript.

Reviewer 1

The manuscript refers to a retrospective analysis of the incidence and mutations of Phenylketonuria in Eastern Sicily. The article is interesting; the rationale is adequate. The techniques used are the standard for genetic diseases. However, the title of the manuscript is too long and confusing, it must be shortened, a suggestion Genotype-phenotype correlation in a large cohort of eastern Sicilian patients: Newborn Screening program, clinical features, and follow-up. The abstract has a similar problem; it must be concise yet informative. Due to the admixed population in Sicily, this reference should be helpful to discuss  and it will interesting to correlate the findings: DOI: 10.1016/j.ebiom.2019.102623

Authors’ Reply: We’ve shortened the abstract and incorporated the title Reviewer 1 suggested; we cited the article with a very large French population, comparing the most common mutations found in our study (Mediterranean France).

  • Please add a Table with the patient's clinical characteristics so it is easier to follow up. In Table 6, please discuss why the responsiveness of BH4 is partial in patient 7 compared to patient 4.

Authors’ Reply: We added a note in the table 6 regarding the difference in response (which could be related to epigenetic modifications or gene modifiers)

  • Table 7 should be only the right side of the Figure.

Authors’ Reply: We changed the order: now table 7 is on the right of the figure

  • Table 8 is confusing; please modify to the short part and add a legend if necessary.

Authors’ Reply: Table 8 has been reformatted and explained in the caption; legend was added

  •  Figure 1: how many individuals are analyzed? The Figure can be simplified by adding age.

Authors’ Reply: We added a caption in the legend explaining the number and the age of the patients

  • In Figure 2, please label the axis and add the correlation value even though none exists.

Authors’ Reply: We changed the figure, adding the label and the numbers in the axis and the pearson value.

  • The discussion should be analyzing the data and comparing it with other studies, not repeating the results and the conclusions, and adding the study's limitations separately.

Authors’ Reply: Conclusion has been rephrased. Limitations have been added to the discussion. Discussion has been rephrased also, avoiding to repeat the conclusions and making it lighter.

Reviewer 2 Report

Comments and Suggestions for Authors

Maria Chiara Consentino et al. reported an interesting work about a patient cohort study. This type of manuscript was less frequently submitted to Nutrients, yet of a certain significance. The manuscript read interesting. Overall, a Minor Revision was suggested for this paper. Please refer to the following comments:

1.      The Abstract contained ~400 words, which was too long to read. Please consider to shorten it to ~200 words to facilitate the rapid reading.

2.      At the end of the Introduction Section, please briefly explain why 102 out of 130 patients were selected in the cohort?

3.      In the Method Section, please consider to move Statistical analysis after Data collection, which was the common style of an Original Article.

4.      The demographic information of the patients could be summarized in the manuscript.

5.      For statistical information shown in Table 8~10: They could be moved to Supplementary Materials.

Author Response

To Prof. Maria Luz Fernandez,

Department of Nutritional Sciences,

University of Connecticut, Storrs, USA

Editor-in-chief

Nutrients

Dear Editor,

We sincerely appreciate the opportunity to revise and resubmit our manuscript (now) entitled Genotype-phenotype correlation in a large cohort of eastern Sicilian patients: Newborn Screening program, clinical features, and follow-up” to Nutrients. We are grateful to the reviewers for their thoughtful and constructive comments, which have greatly helped us improve the quality and clarity of our work.

In this revised version, we have carefully addressed all the points raised by the reviewers and have incorporated their suggestions into the manuscript. Below, we provide a detailed, point-by-point response to each comment, specifying the changes made and referencing the relevant sections of the revised manuscript.

Reviewer 2

Maria Chiara Consentino et al. reported an interesting work about a patient cohort study. This type of manuscript was less frequently submitted to Nutrients, yet of a certain significance. The manuscript read interesting. Overall, a Minor Revision was suggested for this paper. Please refer to the following comments:

  1. The Abstract contained ~400 words, which was too long to read. Please consider to shorten it to ~200 words to facilitate the rapid reading.

Re: It has been shortened into a 200-word abstract

  1. At the end of the Introduction Section, please briefly explain why 102 out of 130 patients were selected in the cohort?

Authors’ Reply: We have changed the last sentence of the introduction accordingly

  1. In the Method Section, please consider to move Statistical analysis after Data collection, which was the common style of an Original Article.

Authors’ Reply: It has been corrected

  1. The demographic information of the patients could be summarized in the manuscript.

Authors’ Reply: The demographic information of the center has now been included in the population study and in the new figure 1.

  1. For statistical information shown in Table 8~10: They could be moved to Supplementary Materials.

Authors’ Reply: Table 8 to 10 have been moved to supplementary table 1 to 3.

We trust that the revisions have adequately addressed the concerns raised and that the manuscript is now suitable for publication in Nutrients.

Thank you for your consideration, and please do not hesitate to contact us if further clarifications are needed.

Prof. Andrea D. Praticò

University Kore of Enna

Enna, Italy

Reviewer 3 Report

Comments and Suggestions for Authors

This study is well-designed and provides useful information with clinical implicaiton. Thus, I just have only minor comments.

1. Please avoid one paragraph with only one sentence, such as the initial introduction section.

2. Please add one figure the show the patients selection.

3. In the method, the authors state "Phe blood levels have been determined by dried blood spots (DBS) and bacterial inhibition assay for patients screened or diagnosed between 1987 and 2011 or by tandem mass spectrometry from 2017 to 2023".  How about the period between 2011 and 2017?

Author Response

To Prof. Maria Luz Fernandez,

Department of Nutritional Sciences,

University of Connecticut, Storrs, USA

Editor-in-chief

Nutrients

Dear Editor,

We sincerely appreciate the opportunity to revise and resubmit our manuscript (now) entitled Genotype-phenotype correlation in a large cohort of eastern Sicilian patients: Newborn Screening program, clinical features, and follow-up” to Nutrients. We are grateful to the reviewers for their thoughtful and constructive comments, which have greatly helped us improve the quality and clarity of our work.

In this revised version, we have carefully addressed all the points raised by the reviewers and have incorporated their suggestions into the manuscript. Below, we provide a detailed, point-by-point response to each comment, specifying the changes made and referencing the relevant sections of the revised manuscript.

Reviewer 3

This study is well-designed and provides useful information with clinical implicaiton. Thus, I just have only minor comments.

  1. Please avoid one paragraph with only one sentence, such as the initial introduction section.

Authors’ Reply: Paragraph formatting has been widely corrected throughout the text.

  1. Please add one figure the show the patients selection.

Authors’ Reply:  The new “Figure 1” reports the full flow-chart of patients selection

  1. In the method, the authors state "Phe blood levels have been determined by dried blood spots (DBS) and bacterial inhibition assay for patients screened or diagnosed between 1987 and 2011 or by tandem mass spectrometry from 2017 to 2023".  How about the period between 2011 and 2017?

Authors’ Reply: It has been corrected (1987-2011 and 2011 to 2023)

We trust that the revisions have adequately addressed the concerns raised and that the manuscript is now suitable for publication in Nutrients.

Thank you for your consideration, and please do not hesitate to contact us if further clarifications are needed.

Prof. Andrea D. Praticò

University Kore of Enna

Enna, Italy

Reviewer 4 Report

Comments and Suggestions for Authors

Dear Authors,

This study was conducted to genotype-phenotype correlation in a large cohort of 102 PKU patients of eastern sicily: newborn screening program, clinical features, genotype, and follow-up of patients. This manuscript has scientific and novel and this topic was uncovered issue from PKU patients. I believe this was excellent issue in field of gene, molecular, medicine, and nursing section.

Title

Please do not use abbreviation and number of participants in Title section. For this reason, please change from 102 PKU patients to phenylketonuria patients in Title.

Abstract

Original research articles should have a structured abstract of around 250 words. Your study consists of over 400 words. Please reduce abstract to around 250 words.

Line 35: 33/102 MHPA à 33/102 mild-hyperphenylalaninemia

Line 40: (BioPKU ) à (BioPKU)

Please add the results of statistical exact p-value in each variables in Results section.

Please sort alphabetically in Key-words.

Introduction

Please could you more explain the purpose of this study for PKU patients. It should be added 3-4 paragraphs for purpose of this study in PKU patients.

Method

Line 159: Microsoft Excel 2016 à Excel in Microsoft® Office 2016 (Microsoft Corporation, Redmond, WA, USA).

Line 160: Jamovi à Jamovi (???, ???, Sydney, Australia).

Line 169: screening management database: please add source internet link.

Results

All Tables and Figures, you have to insert abbreviation and full name in footnote.

Please you should double check name of Genotype in Results section, again. There are too many Genotype variables.

Please revise all results to two decimal places in mean, standard deviation, etc., and three decimal places in statistical values (t, F value, p-value) are generally spelled out in academic writing. Please change in whole manuscript and all Tables.

Discussion

You should add limitation, strengths, and application in field of this study. Moreover, the references are too small numbers. You should add more than 10 references in whole manuscript. Furthermore, checking by the iThenticate system, the plagiarism rate was 17% (quotes included and bibliography excluded). I think it is not acceptable plagiarism rate. Please reduce the plagiarism rate under 15%.

Author Response

To Prof. Maria Luz Fernandez,

Department of Nutritional Sciences,

University of Connecticut, Storrs, USA

Editor-in-chief

Nutrients

Dear Editor,

We sincerely appreciate the opportunity to revise and resubmit our manuscript (now) entitled Genotype-phenotype correlation in a large cohort of eastern Sicilian patients: Newborn Screening program, clinical features, and follow-up” to Nutrients. We are grateful to the reviewers for their thoughtful and constructive comments, which have greatly helped us improve the quality and clarity of our work.

In this revised version, we have carefully addressed all the points raised by the reviewers and have incorporated their suggestions into the manuscript. Below, we provide a detailed, point-by-point response to each comment, specifying the changes made and referencing the relevant sections of the revised manuscript.

Reviewer 4

Dear Authors,

This study was conducted to genotype-phenotype correlation in a large cohort of 102 PKU patients of eastern sicily: newborn screening program, clinical features, genotype, and follow-up of patients. This manuscript has scientific and novel and this topic was uncovered issue from PKU patients. I believe this was excellent issue in field of gene, molecular, medicine, and nursing section.

Title

Please do not use abbreviation and number of participants in Title section. For this reason, please change from 102 PKU patients to phenylketonuria patients in Title.

Authors’ Reply: The title has been changed

Abstract

Original research articles should have a structured abstract of around 250 words. Your study consists of over 400 words. Please reduce abstract to around 250 words.

Authors’ Reply: The abstract has been reduced into a 250-word structured abstract.

Line 35: 33/102 MHPA à 33/102 mild-hyperphenylalaninemia

Authors’ Reply: it has been corrected

 Line 40: (BioPKU ) à (BioPKU)

Authors’ Reply: it has been corrected

Please add the results of statistical exact p-value in each variables in Results section.

Authors’ Reply: in the reduced version of the abstract we’ve reported only the percentage of PKU types and the prevalence of the more common mutations. Such results did not underwent to a statistical analysis (they are not statistical significant). However we added one aspect of the study (dietary tolerance) with the respective pValue comparing the various forms of PKU and m-HPA.

Please sort alphabetically in Key-words.

Authors’ Reply: keywords are in alphabetical order

Introduction

Please could you more explain the purpose of this study for PKU patients. It should be added 3-4 paragraphs for purpose of this study in PKU patients.

Authors’ Reply: we have explained in the last section of the introduction the aim of the present study

Method

Line 159: Microsoft Excel 2016 à Excel in Microsoft® Office 2016 (Microsoft Corporation, Redmond, WA, USA).

Authors’ Reply: it has been corrected

Line 160: Jamovi à Jamovi (???, ???, Sydney, Australia).

Authors’ Reply: it has been corrected

Line 169: screening management database: please add source internet link.

Authors’ Reply: it has been better explained that this is a software

Results

All Tables and Figures, you have to insert abbreviation and full name in footnote.

Authors’ Reply: it has been corrected and abbreviations have been added in footnotes

Please you should double check name of Genotype in Results section, again. There are too many Genotype variables.

Authors’ Reply: we have double-checked all the entries and they correspond to patients results

Please revise all results to two decimal places in mean, standard deviation, etc., and three decimal places in statistical values (t, F value, p-value) are generally spelled out in academic writing. Please change in whole manuscript and all Tables.

Authors’ Reply: all the numbers have been corrected

Discussion

You should add limitation, strengths, and application in field of this study. Moreover, the references are too small numbers. You should add more than 10 references in whole manuscript. Furthermore, checking by the iThenticate system, the plagiarism rate was 17% (quotes included and bibliography excluded). I think it is not acceptable plagiarism rate. Please reduce the plagiarism rate under 15%.

Authors’ Reply: the discussion has been deeply rephrased in order to avoid a plagiarism rate over 15% (now 14%, source: justdone.ai). 10 new references have been added.

We trust that the revisions have adequately addressed the concerns raised and that the manuscript is now suitable for publication in Nutrients.

Thank you for your consideration, and please do not hesitate to contact us if further clarifications are needed.

Prof. Andrea D. Praticò

University Kore of Enna

Enna, Italy

Round 2

Reviewer 1 Report

Comments and Suggestions for Authors

The manuscript was notoriously improved. I have no further comments